# John Ellipsoids via Lazy Updates

**David P. Woodruff**
Carnegie Mellon University
dwoodruf@cs.cmu.edu

**Taisuke Yasuda**
Voleon Group
yasuda.taisuke1@gmail.com

## Abstract

We give a faster algorithm for computing an approximate John ellipsoid around $n$ points in $d$ dimensions. The best known prior algorithms are based on repeatedly computing the leverage scores of the points and reweighting them by these scores [CCLY19]. We show that this algorithm can be substantially sped up by delaying the computation of high accuracy leverage scores by using sampling, and then later computing multiple batches of high accuracy leverage scores via fast rectangular matrix multiplication. We also give low-space streaming algorithms for John ellipsoids using similar ideas.

## 1 Introduction

The *John ellipsoid* problem is a classic algorithmic problem, which takes as input a set of $n$ points $\{\mathbf{a}_1, \mathbf{a}_2, \ldots, \mathbf{a}_n\}$ in $d$ dimensions, and asks for the minimum volume enclosing ellipsoid (MVEE) of these $n$ points. If $P$ is the convex hull of these $n$ points, then a famous result of Fritz John [Joh48] states that such an ellipsoid satisfies $\frac{1}{d}Q \subseteq P \subseteq Q$, and if $P$ is furthermore symmetric, then $\frac{1}{\sqrt{d}}Q \subseteq P \subseteq Q$. Equivalently, we may consider the $n$ input points to be constraints of a polytope $\{\mathbf{x} : \langle \mathbf{a}_i, \mathbf{x} \rangle \leq 1, i \in [n]\}$, in which case the problem is to compute a maximal volume inscribed ellipsoid (MVIE). These two problems are related by taking polars, which corresponds to inverting the quadratic form defining the ellipsoid. In this work, we focus on the symmetric case, so that the polytope $P$ may be written as $P = \{\mathbf{x} : \|\mathbf{A}\mathbf{x}\|_\infty \leq 1\}$, where $\mathbf{A}$ denotes the $n \times d$ matrix with the $n$ input points $\mathbf{a}_i$ in the rows, and our goal is to output an ellipsoid $Q$ which approximately satisfies $Q \subseteq P \subseteq \sqrt{d} \cdot Q$.

The John ellipsoid problem has far-reaching applications in numerous fields of computer science. In statistics, the John ellipsoid problem is equivalent to the dual of the D-optimal experiment design problem [Atw69, Sil13], in which one seeks weights for selecting a subset of a dataset to observe in an experiment. Other statistical applications of John ellipsoids include outlier detection [ST80] and pattern recognition [Ros65, Gli98]. In the optimization literature, computing John ellipsoids is a fundamental ingredient for the ellipsoid method for linear programming [Kha79], cutting plane methods [TKE88], and convex programming [Sho77, Vai96]. Other applications in theoretical computer science include sampling [Vem05, CDWY18, GN23], bandits [BCK12, HK16], differential privacy [NTZ13], coresets [TMF20, TWZ+22], and randomized numerical linear algebra [Cla05, DDH+09, WY23, BMV23].

We refer to a survey of Todd [Tod16] for an account on algorithms and applications of John ellipsoids.

### 1.1 John ellipsoids via iterated leverage scores

Following a long line of work on algorithms for computing John ellipsoids [Wol70, Atw73, KT93, NN94, Kha96, STT78, KY05, SF04, TY07, AST08, Yu11, HFR20] based on convex optimization techniques, the work of [CCLY19] developed a new approach towards computing approximate John

38th Conference on Neural Information Processing Systems (NeurIPS 2024).

ellipsoids via a simple fixed point iteration approach. The approach of [CCLY19] starts with an observation on the optimality condition for the dual problem, given by

$$\text{minimize } \sum_{i=1}^{n} \mathbf{w}_i - \log \det(\mathbf{A}^\top \mathbf{W} \mathbf{A}) - d$$

$$\text{subject to } \mathbf{w}_i \geq 0, i \in [n]$$

where $\mathbf{W} = \operatorname{diag}(\mathbf{w})$.[1] The optimality condition requires that the dual weights $\mathbf{w}$ satisfy

$$\mathbf{w}_i = \boldsymbol{\tau}_i(\sqrt{\mathbf{W}}\mathbf{A}), \tag{1}$$

for each $i \in [n]$, where $\boldsymbol{\tau}_i(\mathbf{A})$ for a matrix $\mathbf{A}$ denotes the $i$-th *leverage score* of $\mathbf{A}$.

**Definition 1.1** (Leverage score). *Let $\mathbf{A} \in \mathbb{R}^{n \times d}$. Then, for each $i \in [n]$, we define the $i$-th leverage score as*

$$\boldsymbol{\tau}_i(\mathbf{A}) := \mathbf{a}_i^\top (\mathbf{A}^\top \mathbf{A})^- \mathbf{a}_i = \sup_{\mathbf{A}\mathbf{x} \neq 0} \frac{[\mathbf{A}\mathbf{x}](i)^2}{\|\mathbf{A}\mathbf{x}\|_2^2}.$$

The optimality condition of (1) can be viewed as a fixed point condition, which suggests the following iterative algorithm for computing approximate John ellipsoids

$$\mathbf{w}_i^{(t)} \leftarrow \boldsymbol{\tau}_i(\sqrt{\mathbf{W}^{(t-1)}}\mathbf{A}) = \mathbf{w}_i^{(t-1)} \cdot \mathbf{a}_i^\top (\mathbf{A}^\top \mathbf{W}^{(t-1)} \mathbf{A})^- \mathbf{a}_i \tag{2}$$

where $\mathbf{W}^{(t)} := \operatorname{diag}(\mathbf{w}^{(t)})$. After repeating this update for $T = O(\varepsilon^{-1} \log(n/d))$ iterations starting with $\mathbf{w}^{(0)} = d/n \cdot 1_n$, it can be shown that the ellipsoid $Q$ defined by the quadratic form $\mathbf{Q} = \mathbf{A}^\top \mathbf{W}^{(T)} \mathbf{A}$, i.e. $Q = \{\mathbf{x} \in \mathbb{R}^d : \mathbf{x}^\top \mathbf{Q} \mathbf{x} \leq 1\}$, satisfies

$$\frac{1}{\sqrt{1+\varepsilon}} Q \subseteq P \subseteq \sqrt{d} \cdot Q. \tag{3}$$

Note that the computation of leverage scores can be done in $O(nd^{\omega-1})$ time, where $\omega \leq 2.371552$ is the current exponent of fast matrix multiplication [DWZ23, WXXZ24]. Thus, this gives an algorithm running in time $O(\varepsilon^{-1} nd^{\omega-1} \log(n/d))$ for outputting an ellipsoid with the guarantee of (3).

It is known that input matrices with additional structure admit even faster implementation of this algorithm. For instance, [CCLY19, SYYZ22] give faster algorithms for sparse matrices $\mathbf{A}$ for which the number of nonzero entries $\operatorname{nnz}(\mathbf{A})$ is much less than $nd$. The work of [SYYZ22] shows that this approach can also be sped up for matrices $\mathbf{A}$ with small treewidth.

## 1.2 Our results

Our first main result is a substantially faster algorithm for computing John ellipsoids. In the typical regime where $n \gg d$, our algorithm runs in $O(\varepsilon^{-1}nd) \log(n/d)$ time to output an ellipsoid $Q$ satisfying (3), and $O(\varepsilon^{-1}nd^2) \log(n/d)$ time to output an ellipsoid $Q$ which approximates the maximal volume up to a $(1 + \varepsilon)$ factor. We will discuss our techniques for this result in Sections 1.2.1 and 1.2.2.

Table 1: Running time of John ellipsoid approximation for dense $n \times d$ matrices, for $n \gg d \gg$ $\operatorname{poly}(\varepsilon^{-1} \log n)$. There is other prior work on sparse matrices and matrices with low treewidth [CCLY19, SYYZ22].

|  | Running time | Guarantee |
| --- | --- | --- |
| [KY05, TY07] | $O(\varepsilon^{-1}nd^{\omega})$ | volume approximation |
| [CCLY19] | $O(\varepsilon^{-1}nd^{\omega-1}) \log(n/d)$ | (3) |
| [CCLY19, SYYZ22] | $O(\varepsilon^{-2}nd)(\log n) \log(n/d)$ | (3) |
| Theorem 1.6 | $O(\varepsilon^{-1}nd) \log(n/d)$ | (3) |

---

[1] For a weight vector $\mathbf{w} \in \mathbb{R}^n$, we will often write the corresponding $n \times n$ diagonal matrix $\operatorname{diag}(\mathbf{w})$ as the capitalized version $\mathbf{W}$.

### 1.2.1 Linear time leverage scores via fast matrix multiplication

We start by showing how to approximate leverage scores up to $(1 + \varepsilon)$ factors in $\tilde{O}(nd)$ time, which had not been observed before to the best of our knowledge. Note that if we compute exact leverage scores using fast matrix multiplication, then this takes time $O(nd^{\omega-1})$. Alternatively, sketching-based algorithms for approximate leverage scores are known, which gives the following running time for sparse matrices $\mathbf{A}$ with $\mathsf{nnz}(\mathbf{A})$ nonzero entries.

**Theorem 1.2** ([SS11, DMMW12, CW13]). *There is an algorithm which, with probability at least $1 - \delta$, outputs $\boldsymbol{\tau}'_i$ for $i \in [n]$ such that*

$$\boldsymbol{\tau}'_i = (1 \pm \varepsilon)\boldsymbol{\tau}_i(\mathbf{A})$$

*and runs in time $O(\varepsilon^{-2}\,\mathsf{nnz}(\mathbf{A})\log(n/\delta)) + \operatorname{poly}(d\varepsilon^{-1}\log(n/\delta))$.*

If the goal is to compute $(1 + \varepsilon)$-approximate leverage scores for a dense $n \times d$ matrix, then we are not aware of a previous result which does this in a nearly linear $\tilde{O}(nd)$ time, which we now show:

**Theorem 1.3.** *There is an algorithm which, with probability at least $1 - \delta$, outputs $\boldsymbol{\tau}'_i$ for $i \in [n]$ such that*

$$\boldsymbol{\tau}'_i = (1 \pm \varepsilon)\boldsymbol{\tau}_i(\mathbf{A})$$

*in time $O(nd)\operatorname{poly}\log(\varepsilon^{-1}\log(n/\delta)) + O(n)\operatorname{poly}(\varepsilon^{-1}\log(n/\delta))$.*

Our improvement comes from improving the running time analysis of a sketching-based algorithm of [DMMW12] by using fast rectangular matrix multiplication. We will need the following result on fast matrix multiplication:

**Theorem 1.4** ([Cop82, Wil11, Wil24]). *There is a constant $\alpha \geq 0.1$ and an algorithm for multiplying a $m \times m$ and a $m \times m^\alpha$ matrix in $O(m^2 \operatorname{poly}\log m)$ time, under the assumption that field operations can be carried out in $O(1)$ time.*

By applying the above result in blocks, we get the following version of this result for rectangular matrix multiplication.

**Corollary 1.5.** *There is a constant $\alpha \geq 0.1$ and an algorithm for multiplying a $n \times d$ for $n \geq d$ and a $d \times t$ matrix in $O(nd + nt^{1/\alpha+1})\operatorname{poly}\log t$ time, under the assumption that field operations can be carried out in $O(1)$ time.*

*Proof.* Let $m = t^{1/\alpha}$. If $d \leq m$, then matrix multiplication takes only $O(ndt) = O(nt^{1/\alpha+1})$ time, so assume that $d \geq m$. We partition the first matrix into an $O(n/m) \times O(d/m)$ block matrix with blocks of size $m \times m$ and the second into an $O(d/m) \times 1$ block matrix with blocks of size $m \times m^\alpha$. By Theorem 1.4, each block matrix multiplication requires $O(m^2 \operatorname{poly}\log m)$ time, and we have $O(nd/m^2)$ of these to do, which gives the desired running time. $\qquad\square$

That is, the above result shows that when multiplying an $n \times d$ matrix $\mathbf{A}$ with a $d \times t$ matrix for a much smaller $t$, then this multiplication can be done in roughly $O(nd)\operatorname{poly}\log t$ time. The work of [DMMW12] shows that the leverage scores of $\mathbf{A}$ can be written as the row norms of $\mathbf{AR}$ for a $d \times t$ matrix with $t = O(\varepsilon^{-2}\log(n/\delta))$, and thus this gives us the result of Theorem 1.3.

### 1.2.2 John ellipsoids via lazy updates

By using Theorem 1.3, we already obtain a John ellipsoid algorithm which runs in time $O(\varepsilon^{-1}nd)\log(n/d)\operatorname{poly}\log(\varepsilon^{-1}\log n)$ time, which substantially improves upon prior algorithms for dense input matrices $\mathbf{A}$. We now show how to obtain further improvements by using the idea of *lazy updates*. At the heart of our idea is to only compute the *quadratic forms* for the John ellipsoids for most iterations, and defer the computation of the weights until we have computed roughly $O(\log n)$ iterations. At the end of this group of iterations, we can then compute the John ellipsoid weights via fast matrix multiplication as used in Theorem 1.3, which allows us to remove the suboptimal $\operatorname{poly}\log\log n$ terms in the dominating term of the running time.

**Theorem 1.6.** *Given $\mathbf{A} \in \mathbb{R}^{n\times d}$, let $P$ be the polytope defined by $P = \{\mathbf{x} \in \mathbb{R}^d : \|\mathbf{Ax}\|_\infty \leq 1\}$. For $\varepsilon \in (0, 1)$, there is an algorithm, Algorithm 3, that runs in time*

$$O(\varepsilon^{-1}nd)(\log(n/d) + \operatorname{poly}\log(\varepsilon^{-1}\log n)) + O(n)\operatorname{poly}(\varepsilon^{-1}\log n) + O(n^{0.1})d^{\omega+1}\varepsilon^{-3}(\log n)^2$$

*and returns an ellipsoid $Q$ such that $\frac{1}{\sqrt{1+\varepsilon}} \cdot Q \subseteq P \subseteq \sqrt{d} \cdot Q$ with probability at least $1 - 1/\operatorname{poly}(n)$.*

The full proof of this result is given in Section 2.

Let $\mathbf{Q}^{(t)} = \mathbf{A}^\top \mathbf{W}^{(t)} \mathbf{A}$, where the weights $\mathbf{w}^{(t)}$ are defined as (2). Note that with this notation, the update rule for the iterative algorithm of [CCLY19] can be written as

$$\mathbf{w}_i^{(t)} = \mathbf{w}_i^{(t-1)} \cdot \mathbf{a}_i^\top (\mathbf{Q}^{(t-1)})^- \mathbf{a}_i. \tag{4}$$

Thus, given high-accuracy spectral estimates to the quadratics $\mathbf{Q}^{(t)}$, we can recover the weights $\mathbf{w}_i^{(t)}$ to high accuracy in $O(d^\omega)$ time per iteration by evaluating the quadratic forms $\mathbf{a}_i^\top (\mathbf{Q}^{(t)})^- \mathbf{a}_i$ and then multiplying them together. This approach is useful for fast algorithms if we only need to to do this for a small number of indices $i \in [n]$. This is indeed the case if we only need these weights for a *row sample* of $\sqrt{\mathbf{W}^{(t)}}\mathbf{A}$, which is sufficient for computing a spectral approximation to the next quadratic form $\mathbf{Q}^{(t)}$. Furthermore, we only need low-accuracy leverage scores (up to a factor of, say, $n^{0.1}$) to obtain a good row sample, which can be done quickly for all $n$ rows [LMP13, CLM$^+$15]. Thus, by repeatedly sampling rows of $\sqrt{\mathbf{W}^{(t)}}\mathbf{A}$, computing high-accuracy weights on the sampled rows, and then building an approximate quadratic, we can iteratively compute high-accuracy approximations $\tilde{\mathbf{Q}}^{(t)}$ to the quadratics $\mathbf{Q}^{(t)}$. More formally, our algorithm takes the following steps:

- We first compute low-accuracy leverage scores of $\sqrt{\mathbf{W}^{(t-1)}}\mathbf{A}$, which can be done in $O(nd)$ time. This gives us the weights $\mathbf{w}^{(t)}$ to low accuracy, say $\mathbf{u}^{(t)}$, for all $n$ rows.

- We use the low-accuracy weights $\mathbf{u}^{(t)}$ to obtain a weighted subset of rows of $\sqrt{\mathbf{U}^{(t)}}\mathbf{A}$ which spectrally approximates $\mathbf{Q}^{(t)}$. Note, however, that we do not yet have the sampled rows of $\sqrt{\mathbf{W}^{(t)}}\mathbf{A}$ to high accuracy, since we do not know the weights $\mathbf{w}^{(t)}$ to high accuracy.

- If we only need the weights $\mathbf{w}^{(t)}$ for a small number of sampled rows $S \subseteq [n]$, then we can explicitly compute these using (4), since we inductively have access to high-accuracy quadratics $\tilde{\mathbf{Q}}^{(t')}$ for $t' = 0, 1, 2, \ldots, t-1$. These can then be used to build $\tilde{\mathbf{Q}}^{(t)}$.

While this algorithm allows us to quickly compute high-accuracy approximate quadratics $\tilde{\mathbf{Q}}^{(t)}$, this algorithm cannot be run for too many iterations, as the error in the low-accuracy leverage scores $\mathbf{u}^{(t)}$ grows to $\mathrm{poly}(n)$ factors in $O(\log n)$ rounds. This is a problem, as this error factor directly influences the number of leverage score samples needed to approximate $\tilde{\mathbf{Q}}^{(t)}$. We will now use the fast matrix multiplication trick from the previous Section 1.2.1 to fix this problem. Indeed, after $O(\log n)$ iterations, we will now have approximate quadratics $\tilde{\mathbf{Q}}^{(1)}, \tilde{\mathbf{Q}}^{(2)}, \ldots, \tilde{\mathbf{Q}}^{(t)}$ for $t = O(\log n)$. Now we just need to compute the $n$ John ellipsoid weights which are given by

$$\mathbf{v}_i^{(t)} = \prod_{t'=1}^{t} \|\mathbf{e}_i^\top \mathbf{A}(\tilde{\mathbf{Q}}^{(t'-1)})^{-1/2}\|_2^2.$$

To approximate this quickly, we can approximate each term $\|\mathbf{e}_i^\top \mathbf{A}(\tilde{\mathbf{Q}}^{(t'-1)})^{-1/2}\|_2^2$ by $\|\mathbf{e}_i^\top \mathbf{A}(\tilde{\mathbf{Q}}^{(t'-1)})^{-1/2}\mathbf{G}^{(t')}\|_2^2$ for a random Gaussian matrix $\mathbf{G}^{(t')}$, by the Johnson–Lindenstrauss lemma [JL84]. Here, the number of columns of the Gaussian matrix can be taken to be $\mathrm{poly}(\varepsilon^{-1} \log n)$, so now all we need to compute is the matrix product

$$\mathbf{A} \cdot [(\tilde{\mathbf{Q}}^{(0)})^{-1/2}\mathbf{G}^{(0)}, (\tilde{\mathbf{Q}}^{(1)})^{-1/2}\mathbf{G}^{(1)}, \ldots, (\tilde{\mathbf{Q}}^{(t)})^{-1/2}\mathbf{G}^{(t)}]$$

which is the product of a $n \times d$ matrix and a $d \times m$ matrix for $m = \mathrm{poly}(\varepsilon^{-1} \log n)$. By Theorem 1.4, this can be computed in $O(nd \, \mathrm{poly} \log m)$ time. However, this resetting procedure is only run $O(\varepsilon^{-1})$ times across the $T = O(\varepsilon^{-1} \log n)$ iterations, so the running time contribution from the resetting is just

$$O(\varepsilon^{-1}nd) \, \mathrm{poly} \log(\varepsilon^{-1} \log n) + O(n) \, \mathrm{poly}(\varepsilon^{-1} \log n).$$

Overall, the total running time of our algorithm is

$$O(\varepsilon^{-1}nd)(\log(n/d) + \mathrm{poly} \log(\varepsilon^{-1} \log n)) + O(n) \, \mathrm{poly}(\varepsilon^{-1} \log n).$$

**Remark 1.7.** *In general, our techniques can be be viewed as a way of exploiting the increased efficiency of matrix multiplication when performed on a larger instance by delaying large matrix multiplication operations, so that the running time is $O(\varepsilon^{-1}nd \log(n/d)) + O(\varepsilon^{-1})T_r$ where $T_r$ is the time that it takes to multiply $\mathbf{A}$ by a $d \times r$ matrix for $r = \mathrm{poly}(\varepsilon^{-1} \log n)$. While we have instantiated this general theme by obtaining faster running times via fast matrix multiplication, one can expect similar improvements by other ways of exploiting the economies of scale of matrix multiplication, such as parallelization. For instance, we recover the same running time if we can multiply $r$ vectors in parallel so that $T_r = O(nd)$.*

### 1.2.3 Streaming algorithms

The problem of computing John ellipsoids is also well-studied in the *streaming model*, where the input points $\mathbf{a}_i$ arrive one at a time in a stream [MSS10, AS15, WY22, MMO22, MMO23]. The streaming model is often considered when the number of points $n$ is so large that we cannot fit all of the points in memory at once, and the focus is primarily on designing algorithms with low space complexity. Our second result of this work is that approaches similar to the one we take to prove Theorem 1.6 in fact also give a low-space implementation of the iterative John ellipsoid algorithm of [CCLY19].

**Theorem 1.8** (Streaming algorithms). *Given $\mathbf{A} \in \mathbb{R}^{n \times d}$, let $P$ be the polytope defined by $P = \{\mathbf{x} \in \mathbb{R}^d : \|\mathbf{A}\mathbf{x}\|_\infty \leq 1\}$. Furthermore, suppose that $\mathbf{A}$ is presented in a stream where the rows $\mathbf{a}_i \in \mathbb{R}^d$ arrive one by one in a stream. For $\varepsilon \in (0,1)$, there is an algorithm, Algorithm 1, that makes $T = O(\varepsilon^{-1} \log(n/d))$ passes over the stream, takes $O(d^2 T)$ time to update after each new row, and returns an ellipsoid $Q$ such that $\frac{1}{\sqrt{1+\varepsilon}} \cdot Q \subseteq P \subseteq \sqrt{d} \cdot Q$. Furthermore, the algorithm uses at most $O(d^2 T)$ words of space.*

In Section 1.2.2, we showed that by storing only the quadratics $\tilde{\mathbf{Q}}^{(t)}$ and only computing the weights $\prod_{t'=1}^{t} \mathbf{a}_i (\tilde{\mathbf{Q}}^{(t'-1)})^{-} \mathbf{a}_i$ as needed, we could design fast algorithms for John ellipsoids. In fact, this idea is also useful in the streaming setting, since storing all of the weights $\mathbf{w}_i^{(t)}$ requires $O(n)$ space per iteration, whereas storing the quadratics $\tilde{\mathbf{Q}}^{(t)}$ requires only $O(d^2)$ space per iteration. Furthermore, in the streaming setting, we may optimize the update running time by instead storing the pseudo-inverse of the quadratics $(\tilde{\mathbf{Q}}^{(t)})^{-}$ and then updating them by using the Sherman-Morrison low rank update formula.[2] This gives the result of Theorem 1.8.

---

**Algorithm 1** Streaming John ellipsoids via lazy updates

---

1: **function** STREAMINGJOHNELLIPSOID(input matrix $\mathbf{A}$)
2:     **for** $t = 0$ to $T$ **do**
3:         Let $\mathbf{Q}^{(t)} = 0$
4:         **for** $i = 1$ to $n$ **do**
5:             **if** $t = 0$ **then**
6:                 Let $\mathbf{Q}^{(t)} \leftarrow \mathbf{Q}^{(t)} + \mathbf{a}_i \mathbf{a}_i^\top$
7:             **else**
8:                 Let $\mathbf{w}_i^{(t)} = \prod_{t'=1}^{t} \mathbf{a}_i^\top (\mathbf{Q}^{(t'-1)})^{-} \mathbf{a}_i$
9:                 Let $\mathbf{Q}^{(t)} \leftarrow \mathbf{Q}^{(t)} + \mathbf{w}_i^{(t)} \mathbf{a}_i \mathbf{a}_i^\top$
10:     **return** $\frac{1}{T+1} \sum_{t=0}^{T} \mathbf{Q}^{(t)}$

---

### 1.3 Notation

Throughout this paper, $\mathbf{A}$ will denote an $n \times d$ matrix whose $n$ rows are given by vectors $\mathbf{a}_i \in \mathbb{R}^d$. For positive numbers $a, b > 0$, we write $a = (1 \pm \varepsilon)b$ to mean that $(1-\varepsilon)b \leq a \leq (1+\varepsilon)b$. For symmetric positive semidefinite matrices $\mathbf{Q}, \mathbf{R}$, we write $\mathbf{Q} = (1 \pm \varepsilon)\mathbf{R}$ to mean that $(1-\varepsilon)\mathbf{R} \preceq \mathbf{Q} \preceq (1+\varepsilon)\mathbf{R}$, where $\preceq$ denotes the Löwner order on PSD matrices.

## 2 Fast algorithms

### 2.1 Approximating the quadratics

We will analyze the following algorithm, Algorithm 2, for approximating the quadratics $\mathbf{Q}^{(t)}$ of the iterative John ellipsoid algorithm.

Fix a row $i$. Note then that for each iteration $t$, $\|\mathbf{e}_i^\top \mathbf{A}(\tilde{\mathbf{Q}}^{(t-1)})^{-1/2} \mathbf{G}\|_2^2$ is distributed as an independent $\chi^2$ variable with $k$ degrees of freedom, scaled by $\|\mathbf{e}_i^\top \mathbf{A}(\tilde{\mathbf{Q}}^{(t-1)})^{-1/2}\|_2^2$, say $X_t$. Note then that

---

[2] We note that storing the pseudo-inverse may increase the space complexity by polynomial factors in the bit complexity model.

---
**Algorithm 2** Approximating the quadratics
---
1: **function** APPROXQUADRATIC(input matrix $\mathbf{A}$, initial weights $\tilde{\mathbf{w}}^{(0)}$, number of rounds $T$)
2:     Let $\tilde{\mathbf{Q}}^{(0)} = \mathbf{A}^\top \tilde{\mathbf{W}}^{(0)} \mathbf{A}$ for $\tilde{\mathbf{W}}^{(0)} = \operatorname{diag}(\tilde{\mathbf{w}}^{(0)})$.
3:     **for** $t = 1$ to $T$ **do**
4:         Compute $\mathbf{A}(\tilde{\mathbf{Q}}^{(t-1)})^{-1/2}\mathbf{G}$ for a $d \times k$ Gaussian matrix $\mathbf{G}$.
5:         Let $\mathbf{u}_i^{(t)} = \mathbf{u}_i^{(t-1)} \cdot \|\mathbf{e}_i^\top \mathbf{A}(\tilde{\mathbf{Q}}^{(t-1)})^{-1/2}\mathbf{G}\|_2^2$ for each $i \in [n]$.
6:         Let $\mathbf{S}^{(t)}$ be a $(1+\varepsilon)$-approximate leverage score sample of $\sqrt{\mathbf{U}^{(t)}}\mathbf{A}$ (Thm. 2.3).
7:         For rows $i$ sampled by $\mathbf{S}^{(t)}$, set $\mathbf{v}_i^{(t)} = \prod_{t'=1}^{t} \mathbf{a}_i^\top (\tilde{\mathbf{Q}}^{(t'-1)})^{-} \mathbf{a}_i$.
8:         Let $\tilde{\mathbf{Q}}^{(t)} = (\mathbf{S}^{(t)} \sqrt{\mathbf{V}^{(t)}} \mathbf{A})^\top \mathbf{S}^{(t)} \sqrt{\mathbf{V}^{(t)}} \mathbf{A}$.
9:     **return** $\{\tilde{\mathbf{Q}}^{(t)}\}_{t=0}^{T}$
---

after $T$ iterations,

$$\mathbf{u}_i^{(T)} = \prod_{t=1}^{T} \|\mathbf{e}_i^\top \mathbf{A}(\tilde{\mathbf{Q}}^{(t-1)})^{-1/2}\mathbf{G}\|_2^2,$$

which is distributed as

$$\mathbf{u}_i^{(T)} \sim \prod_{t=1}^{T} \|\mathbf{e}_i^\top \mathbf{A}(\tilde{\mathbf{Q}}^{(t-1)})^{-1/2}\|_2^2 \cdot X_t = \prod_{t=1}^{T} \|\mathbf{e}_i^\top \mathbf{A}(\tilde{\mathbf{Q}}^{(t-1)})^{-1/2}\|_2^2 \cdot \prod_{t=1}^{T} X_t \qquad (5)$$

for i.i.d. $\chi^2$ variables $X_t$ with $k$ degrees of freedom. We will now bound each of the terms in this product.

### 2.1.1  Bounds on products of $\chi^2$ variables

We need the following bound on products of $\chi^2$ variables, which generalizes [SSK17, Proposition 1].

**Lemma 2.1.** *Let $X_1, X_2, \ldots, X_t$ be $t$ i.i.d. $\chi^2$ variables with $k$ degrees of freedom. Then,*

$$\mathbf{Pr}\left\{ \prod_{i=1}^{t} X_i \leq \frac{1}{R} \right\} \leq \inf_{s \in (0, k/2)} C_{-s,k}^t R^{-s}$$

*and*

$$\mathbf{Pr}\left\{ \prod_{i=1}^{t} X_i \geq R \right\} \leq \inf_{s > -k/2} C_{s,k}^t R^{-s}$$

*where*

$$C_{s,k} = \frac{2^s \Gamma(s + k/2)}{\Gamma(k/2)} > 0$$

*Proof.* For $s > -k/2$, the moment generating function of $\log X_i$ is given by

$$\mathbf{E}\, e^{s \log X_i} = \mathbf{E}\, X_i^s = \frac{1}{2^{k/2}\Gamma(k/2)} \int_0^\infty x^s x^{k/2-1} e^{-x/2}\, dx = \frac{2^s \Gamma(s+k/2)}{\Gamma(k/2)} = C_{s,k}.$$

Then by Chernoff bounds,

$$\mathbf{Pr}\left\{ \prod_{i=1}^{t} X_i \leq \frac{1}{R} \right\} = \mathbf{Pr}\left\{ \exp\left( \sum_{i=1}^{t} -s \log X_i \right) \geq R^s \right\} \leq \mathbf{E}[e^{-s \log X_i}]^t R^{-s} = C_{-s,k}^t R^{-s}$$

and

$$\mathbf{Pr}\left\{ \prod_{i=1}^{t} X_i \geq R \right\} = \mathbf{Pr}\left\{ \exp\left( \sum_{i=1}^{t} s \log X_i \right) \geq R^s \right\} \leq \mathbf{E}[e^{s \log X_i}]^t R^{-s} = C_{s,k}^t R^{-s}$$

$\square$

Using the above result, we can show that if the $\chi^2$ variables have $k = O(1/\theta)$ degrees of freedom and the number of rounds $T$ is $c \log n$ for a small enough constant $c$, then the product of the $\chi^2$ variables $\prod_{t=1}^{T} X_t$ will be within a $n^\theta$ factor for some small constant $\theta > 0$.

**Lemma 2.2.** *Fix a small constant $\theta > 0$ and let $k = O(1/\theta)$. Let $T = c \log n$ for a sufficiently small constant $c > 0$. Let $X_1, X_2, \ldots, X_T$ be $t$ i.i.d. $\chi^2$ variables with $k$ degrees of freedom. Then,*

$$\mathbf{Pr}\left\{ \frac{1}{n^\theta} \leq \prod_{i=1}^{t} X_i \leq n^\theta \right\} \geq 1 - \frac{1}{\mathrm{poly}(n)}.$$

*Proof.* Set $s = k/2$. Then, $C_{-s,k}$ and $C_{s,k}$ are absolute constants. Now let $c$ to be small enough (depending on $s$ and $k$) such that for $T = c \log n$, $C_{-s,k}^T, C_{s,k}^T \leq n$. Furthermore, set $R = n^\theta$. Then, by Lemma 2.1, we have that both $\mathbf{Pr}\{\prod_{i=1}^{t} X_i \leq \frac{1}{R}\}$ and $\mathbf{Pr}\{\prod_{i=1}^{t} X_i \geq R\}$ are bounded by $n \cdot R^{-s} = n \cdot n^{\theta \cdot s} = 1/\mathrm{poly}(n)$, as desired. $\qquad \square$

It follows that with probability at least $1 - 1/\mathrm{poly}(n)$, the products of $\chi^2$ variables appearing in (5) are bounded by $n^\theta$ for every row $i \in [n]$ and for every iteration $t \in [T]$. We will condition on this event in the following discussion.

### 2.1.2 Bounds on the quadratic $\tilde{\mathbf{Q}}^{(t)}$

In the previous section, we have established that the products of $\chi^2$ variables in (5) are bounded by $n^\theta$. We will now use this fact to show that the quadratics $\tilde{\mathbf{Q}}^{(t-1)}$ in Algorithm 2 are good approximate John ellipsoid quadratics. We will use the following leverage score sampling theorem for this.

**Theorem 2.3** ([DMM06, SS11]). *Let $\mathbf{A} \in \mathbb{R}^{n \times d}$. Let $\tau_i' \geq \tau_i(\mathbf{A})$ and let $p_i = \min\{1, \tau_i'/\alpha\}$ for $\alpha = \Theta(\varepsilon^2)/\log(d/\delta)$. If $\mathbf{S}$ is a diagonal matrix with $\mathbf{S}_{i,i} = 1/\sqrt{p_i}$ with probability $p_i$ and $0$ otherwise for $i \in [n]$, then with probability at least $1 - \delta$,*

$$\text{for all } \mathbf{x} \in \mathbb{R}^d, \qquad \|\mathbf{S}\mathbf{A}\mathbf{x}\|_2^2 = (1 \pm \varepsilon)\|\mathbf{A}\mathbf{x}\|_2^2.$$

This theorem, combined with the bounds on $\chi^2$ products, gives the following guarantee for the approximate quadratics $\tilde{\mathbf{Q}}^{(t)}$.

**Lemma 2.4.** *Fix a small constant $\theta > 0$ and let $k = O(1/\theta)$. Suppose that the leverage score sample in Line 6 oversamples by a factor of $O(n^{2\theta})$, that is, uses leverage score estimates $\tau_i'$ such that $\tau_i' \geq O(n^{2\theta})\tau_i(\sqrt{\mathbf{U}^{(t)}}\mathbf{A})$. Then, with probability at least $1 - 1/\mathrm{poly}(n)$, we have for every $t \in [T]$ that*

$$\tilde{\mathbf{Q}}^{(t)} = (1 \pm \varepsilon)\mathbf{A}^\top \mathbf{V}^{(t)} \mathbf{A} \tag{6}$$

*where $\mathbf{v}_i^{(t)} = \prod_{t'=1}^{t-1} \mathbf{a}_i^\top (\tilde{\mathbf{Q}}^{(t')})^- \mathbf{a}_i$ for $i \in [n]$. Furthermore, $\tilde{\mathbf{Q}}^{(t)}$ can be computed in $O(n^{2\theta})T\varepsilon^{-2}d^{\omega+1}\log n$ time in each iteration.*

*Proof.* We will first condition on the success of the event of Lemma 2.2, so that the $\chi^2$ products in (5) are bounded by $n^\theta$ factors for every row $i \in [n]$ and every iteration $t \in [t]$. We will also condition on the success of the leverage score sampling for all $T$ iterations.

Note that by (5) and the bound the $\chi^2$ products, $\mathbf{u}_i^{(t)}$ is within a $O(n^{2\theta})$ factor of $\mathbf{v}_i^{(t)}$, and thus $\mathbf{S}^{(t)}$ is a correct leverage score sample for $\sqrt{\mathbf{V}^{(t)}}\mathbf{A}$. We thus have that

$$\tilde{\mathbf{Q}}^{(t)} = (\mathbf{S}^{(t)}\sqrt{\mathbf{V}^{(t)}}\mathbf{A})^\top \mathbf{S}^{(t)}\sqrt{\mathbf{V}^{(t)}}\mathbf{A} = (1 \pm \varepsilon)\mathbf{A}^\top \mathbf{V}^{(t)} \mathbf{A}.$$

For the running time, note that $\mathbf{S}^{(t)}$ samples at most $O(\varepsilon^{-2}n^{2\theta}d\log n)$ rows in each iteration, and each sampled row $i$ requires $O(d^\omega T)$ to compute $\mathbf{v}_i^{(t)}$. This gives the running time claim. $\qquad \square$

---

**Algorithm 3** John ellipsoids via lazy updates

---

1: **function** JOHNELLIPSOID(input matrix $\mathbf{A}$)
2:      Let $B = O(c^{-1}\varepsilon^{-1})$ and $T = O(c \log(n/d))$ for a sufficiently small constant $c$.
3:      Let $\tilde{\mathbf{w}}_i^{(0)} = d/n$ for $i \in [n]$.
4:      **for** $b = 1$ to $B$ **do**
5:           Let $\{\tilde{\mathbf{Q}}^{(t)}\}_{t=0}^T$ be given by APPROXQUADRATIC($\mathbf{A}, \tilde{\mathbf{w}}^{(0)}$) (Algorithm 2).
6:           Let $\mathbf{G}^{(t-1)}$ for $t \in [T]$ be a random $d \times m$ Gaussian for $m = O(\varepsilon^{-2}(BT)^2 \log n)$.
7:           Compute $\mathbf{A} \cdot [(\tilde{\mathbf{Q}}^{(0)})^{-1/2}\mathbf{G}^{(0)}, (\tilde{\mathbf{Q}}^{(1)})^{-1/2}\mathbf{G}^{(1)}, \ldots, (\tilde{\mathbf{Q}}^{(T)})^{-1/2}\mathbf{G}^{(T)}]$.
8:           Let $\tilde{\mathbf{w}}_i^{(b,t)} = \prod_{t'=1}^t \frac{1}{m} \|\mathbf{e}_i^\top \mathbf{A}(\tilde{\mathbf{Q}}^{(t'-1)})^{-1/2}\mathbf{G}^{(t'-1)}\|_2^2$ for each $i \in [n]$.
9:           Let $\tilde{\mathbf{w}}^{(0)} = \tilde{\mathbf{w}}^{(b,T)}$.
10:     Let $\tilde{\mathbf{w}} = \frac{1}{BT} \sum_{b,t} \tilde{\mathbf{w}}^{(b,t)}$.
11:     **return** $\mathbf{A}^\top \tilde{\mathbf{W}} \mathbf{A}$

---

## 2.2 Full algorithm

We now combine the subroutine for approximating quadratics from Section 2.1 with a resetting procedure using fast matrix multiplication to obtain our full algorithm for quickly computing John ellipsoids. The full algorithm is presented in Algorithm 3.

We will need the following theorem, which summarizes results of [CCLY19] on guarantees of the fixed point iteration algorithm (2) under approximate leverage score computations.

**Theorem 2.5** (Lemma 2.3 and Lemma C.4 of [CCLY19]). *Let $\mathbf{A} \in \mathbb{R}^{n \times d}$ and let $P = \{\mathbf{x} : \|\mathbf{A}\mathbf{x}\|_\infty \leq 1\}$. Let $T = O(\varepsilon^{-1} \log(n/d))$. Suppose that*

$$\mathbf{w}_i^{(t)} = (1 \pm \varepsilon)\boldsymbol{\tau}_i(\sqrt{\mathbf{W}^{(t-1)}}\mathbf{A})$$

*for all $t \in [T]$ and $i \in [n]$. Then, if $Q$ is the ellipsoid given by the quadratic form $\mathbf{A}^\top \mathbf{W}^{(T)}\mathbf{A}$, then $\frac{1}{\sqrt{1+\varepsilon}} \cdot Q \subseteq P \subseteq \sqrt{d} \cdot Q$.*

We also use the Johnson–Lindenstrauss lemma, which is a standard tool for randomized numerical linear algebra, especially in the context of approximating leverage scores [SS11, DMMW12, LMP13, CLM+15].

**Theorem 2.6** ([JL84, DG03]). *Let $m = O(\varepsilon^{-2} \log(n/\delta))$ and let $\mathbf{G}$ be a random $m \times d$ Gaussian matrix. Then, for any $n$ points $\mathbf{a}_1, \mathbf{a}_2, \ldots, \mathbf{a}_n \in \mathbb{R}^d$ in $d$ dimensions, we have with probability at least $1 - \delta$ that*

$$\frac{1}{m}\|\mathbf{G}\mathbf{a}_i\|_2^2 = (1 \pm \varepsilon)\|\mathbf{a}_i\|_2^2$$

*simultaneously for every $i \in [n]$.*

We will now give a proof of Theorem 1.6.

*Proof of Theorem 1.6.* We first argue the correctness of this algorithm. By Theorem 2.5, it suffices to verify that $\tilde{\mathbf{w}}_i^{(b,t)}$ computes $(1 + \varepsilon)$-approximate leverage scores in order to prove the claimed guarantees of Theorem 1.6. For the updates within APPROXQUADRATIC (Algorithm 2), we have by Lemma 2.4 that

$$\tilde{\mathbf{Q}}^{(t)} = (1 \pm \varepsilon)\mathbf{A}^\top \mathbf{V}^{(t)}\mathbf{A}.$$

Thus,

$$\mathbf{v}_i^{(t-1)} \cdot \|\mathbf{e}_i^\top \mathbf{A}(\tilde{\mathbf{Q}}^{(t-1)})^{-1/2}\|_2^2 = (1 \pm \varepsilon)\mathbf{v}_i^{(t-1)} \cdot \|\mathbf{e}_i^\top \mathbf{A}(\mathbf{A}^\top \mathbf{V}^{(t-1)}\mathbf{A})^{-1/2}\|_2^2$$

$$= (1 \pm \varepsilon)\boldsymbol{\tau}_i(\sqrt{\mathbf{V}^{(t-1)}}\mathbf{A})$$

and thus the weights $\mathbf{v}_i^{(t)}$ output by APPROXQUADRATIC (Algorithm 2) indeed satisfy the requirements of Theorem 2.5.

For the weights computed in Line 8 of Algorithm 3, note that we also compute these weights, but this time with approximation error from the application of the Gaussian matrix $\frac{1}{m}\mathbf{G}^{(t)}$ to speed up the

computation. Applying Theorem 2.6 with $\varepsilon$ set to $\varepsilon/BT$ and failure probability $\delta = 1/\operatorname{poly}(n)$, we have that

$$\frac{1}{m}\|\mathbf{e}_i^\top \mathbf{A}(\tilde{\mathbf{Q}}^{(t'-1)})^{-1/2}\mathbf{G}^{(t'-1)}\|_2^2 = (1 \pm \varepsilon/BT)\|\mathbf{e}_i^\top \mathbf{A}(\tilde{\mathbf{Q}}^{(t'-1)})^{-1/2}\|_2^2.$$

Note then that Line 8 takes a product of at most $BT$ of these approximations, so the total error in the approximation is at most

$$(1 \pm \varepsilon/BT)^{BT} = (1 \pm O(\varepsilon)).$$

The running time is given by $BT$ iterations of the inner loop of APPROXQUADRATIC and $B$ iterations of the fast matrix multiplication procedure in Line 7 of Algorithm 3. The inner loop of APPROXQUADRATIC requires $O(nd)$ time to compute the product with the $d \times k$ Gaussian as well as the time to compute the approximate quadratic, which is bounded in Lemma 2.4. Altogether, this gives the claimed running time bound. □

## 3 Future directions

In this work, we developed fast algorithms and low-space streaming algorithms for the problem of computing John ellipsoids. Our fast algorithms use a combination of using lazy updates together with fast matrix multiplication to substantially improve the running time of John ellipsoids, and we apply similar ideas to obtain a low-space streaming implementation of the John ellipsoid algorithm.

Our results have several limitations that we discuss here, which we leave for future work to resolve. First, our algorithm makes crucial use of fast matrix multiplication in order to get running time improvements. However, this makes it a less attractive option for practical implementations, and also makes the polynomial dependence on $\varepsilon$ in the $\tilde{O}(n)\operatorname{poly}(\varepsilon^{-1})$ term rather large. Thus, it is an interesting question whether the running time that we obtain in Theorem 1.6 is possible without fast matrix multiplication.

**Question 3.1.** *Is there an algorithm with the guarantee of Theorem 1.6 that avoids fast matrix multiplication?*

More generally, it is an interesting question to design algorithms for approximating John ellipsoids with optimal running time. This is an old question which has been studied in a long line of work [Wol70, Atw73, KT93, NN94, Kha96, STT78, KY05, SF04, TY07, AST08, Yu11, HFR20], and we believe that the investigation of this question will lead to further interesting developments in algorithms research.

**Question 3.2.** *What is the optimal running time of approximating John ellipsoids?*

For instance, one interesting question is whether it is possible to obtain *nearly linear time* algorithms that run in time $\tilde{O}(nd) + \tilde{O}(n)\operatorname{poly}(\varepsilon^{-1})$, or even *input sparsity time algorithms* that run in time $\tilde{O}(\operatorname{nnz}(\mathbf{A})) + \tilde{O}(n)\operatorname{poly}(\varepsilon^{-1})$. The resolution of such questions for the least squares linear regression problem has led to great progress in algorithms, and studying these questions for the John ellipsoid problem may have interesting consequences as well.

John ellipsoids are closely related to $\ell_p$ *Lewis weights*, which give a natural $\ell_p$ generalization of John ellipsoids and leverage scores and have been a valuable tool in randomized numerical linear algebra. There has been a recent focus on fast algorithms for computing $(1 + \varepsilon)$-approximate $\ell_p$ Lewis weights [FLPS22, AGS24], and thus it is natural to ask whether developments in algorithms for John ellipsoids would carry over to algorithms for $\ell_p$ Lewis weights as well.

**Question 3.3.** *What is the optimal running time of approximating $\ell_p$ Lewis weights?*

Finally, we raise questions concerning streaming algorithms for approximating John ellipsoids. In Theorem 1.8, we gave a multi-pass streaming algorithm which obtains a $(1+\varepsilon)$-optimal John ellipsoid. A natural question is whether a similar algorithm can be achieved in fewer passes, or if there are pass lower bounds for computing $(1 + \varepsilon)$-optimal John ellipsoids.

**Question 3.4.** *Are there small space streaming algorithms for approximating John ellipsoids up to a $(1 + \varepsilon)$ factor which make fewer than $O(\varepsilon^{-1}\log(n/d))$ passes, or do small space streaming algorithms necessarily require many passes?*

We note that there are one-pass small space streaming algorithms if we allow for approximation factors that scale as $O(\sqrt{\log n})$ rather than $(1 + \varepsilon)$ [WY22, MMO22, MMO23].

## Acknowledgments and Disclosure of Funding

The authors were supported in part by a Simons Investigator Award and NSF CCF-2335412.

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
