# OpenReview forum: "John Ellipsoids via Lazy Updates"
_NeurIPS.cc/2024/Conference — NeurIPS 2024 poster_

### Official Review · Reviewer_wsbb · 2024-06-13

**Soundness:** 2
**Presentation:** 2
**Contribution:** 2
**Rating:** 6
**Confidence:** 3

**Summary:**

The paper proposes provably faster algorithm for computing John ellipsoids. The first idea is to approximate Leverage score with an existing faster algorithm given by Theorem 1.3. The second idea is to use fast matrix multiplication and lazy updates to reduce polylog suboptimality.

**Strengths:**

- Provably faster algorithm proposed

**Weaknesses:**

- despite that the main result does not rely on Theorem 1.3, there is no proof or reference for it.
- contribution should clarified. The proof of Theorem 1.6 mainly relies on the existing results.

**Questions:**

Could you please clarify on your particular contribution?

Could you please refer to the proof of Theorem 1.3?

I am not an expert in John Ellipsoids. For me it seemed that the contribution was incremental, because the theoretical result combines existing techniques. By the way, the new result is obtained.

I would also like you to comment on the most recent papers and preprints from 2023 and 2024 if any, to confirm your contribution.

---

> ### Author Rebuttal · Authors · 2024-08-06
>
> > despite that the main result does not rely on Theorem 1.3, there is no proof or reference for it.
>
> > Could you please refer to the proof of Theorem 1.3?
>
> The proof of Theorem 1.3 is sketched in the paragraphs immediately below it.
>
> We first summarize the result of [DMMW12]. It is known that we can compute $Q$ such that $(1+\epsilon)Q \preceq A^\top A \preceq (1-\epsilon) Q$ in time $\mathrm{nnz}(A) + \mathrm{poly}(d/\epsilon)$ time [CW13], so the leverage scores of $A$ are the row norms of $A Q^{-1/2}$ up to a $(1\pm\epsilon)$ factor. We can then apply the Johnson--Lindenstrauss lemma and multiply this matrix by a random projection matrix $P$ with $t = O(\epsilon^{-2}\log n)$ rows, so that the row norms of $A Q^{-1/2} P$ approximate the row norms of $A Q^{-1/2}$ up to a $(1\pm\epsilon)$ factor. Thus, we indeed have that the leverage scores are approximated by the row norms of some matrix $AR$ for $R = Q^{-1/2}P$ up to $(1\pm\epsilon)$ factors.
>
> Finally, we can efficiently multiply $A$ by $Q^{-1/2}P$ quickly using Corollary 1.5. In the revision, we will include a full proof in the appendix.
>
> > contribution should clarified. The proof of Theorem 1.6 mainly relies on the existing results.
>
> > Could you please clarify on your particular contribution?
>
> Our result gives the fastest known algorithm for computing $(1+\epsilon)$-approximate John ellipsoids when $\epsilon$ is not too small. While we indeed build on existing techniques, and in particular the iterated leverage score algorithm of [CCLY19], we make two key improvements to this algorithm: (1) the use of lazy updates that allow us to better exploit the faster running time of leverage scores by batching the leverage score computations, and (2) the use of fast matrix multiplication to give a faster algorithm for approximating leverage scores. Finally, these ideas also give low-space streaming algorithms.
>
> > I would also like you to comment on the most recent papers and preprints from 2023 and 2024 if any, to confirm your contribution.
>
> To the best of our knowledge, the most recent relevant work on computing $(1+\epsilon)$-approximate John ellipsoids is the works of [CCLY22] and [SYYZ22], which are discussed thoroughly in the introduction. We are not aware of works from 2023 or 2024 on this topic.

---

### Official Review · Reviewer_VRzA · 2024-07-10

**Soundness:** 4
**Presentation:** 3
**Contribution:** 3
**Rating:** 6
**Confidence:** 4

**Summary:**

The authors give a leverage score approximation algorithm that runs in nearly linear time for dense matrices. Specifically, to find leverage scores for a matrix $A \in \mathbb{R}^{n \times d}$, they give an algorithm based on fast matrix multiplication that runs in time $\widetilde{O}(nd)$. This, combined with a "lazy update" trick, yield a new fast algorithm to approximate the John ellipsoid for a symmetric convex polyhedron represented as the set $\{x \in \mathbb{R}^d \colon \| Ax \|_{\infty} \le 1\}$. The same ideas also transfer to a $\sim \log n$-pass streaming setting to find the John ellipsoid, yielding a new low space complexity algorithm for finding the John ellipsoid that has a stronger approximation guarantee than existing one-pass solutions.

**Strengths:**

The leverage score approximation algorithm the authors give may yield faster algorithms for a class of optimization algorithms that use leverage score computations as a primitive (one example I can think of offhand is the work [JLS21]). In particular, this should imply a $\widetilde{O}(nd)$ time algorithm for finding weights $w$ such that $2w_i \ge \tau_i(W^{1/2-1/p}A)$ for $p \ge 2$ (which is a one-sided Lewis weight approximation that suffices for a large number of applications, including $\ell_p$ row sampling and $\ell_p$ regression for $p \ge 2$). The insight is pretty simple and cleanly presented, which I view as a positive. I suspect a similar result is true (along with a corresponding low-space streaming algorithm) for for $p < 2$ by applying this to the natural contraction mapping that computes the weights.

[JLS21] Improved Iteration Complexities for Overconstrained p-Norm Regression (https://arxiv.org/abs/2111.01848)

**Weaknesses:**

I wish more applications had been discussed. Leverage scores are a pretty fundamental primitive that get used in a lot of problems in optimization and numerical linear algebra, and it would have been nice to write down concrete runtime improvements (if any) that emerge from Theorem 1.3. I alluded to some in the previous section, and I would be happy to raise my score if the authors either confirm the above or present a few more settings in which concrete runtime improvements are realized. Alternately, I'd also raise the score if the authors discussed a bit about the barriers behind extending their approach to these settings.

Mild issues:

I think there are typos in Algorithm 1 and a suboptimal runtime guarantee. I think Line 8 should read:

$w_{i}^{(t)} = \prod_{t'=1}^{t} a_i^{\top}(A^{\top}W^{(t')}A)^{-1}a_i$. In particular, as written, Line 8 multiplies across the first $t$ rows for a fixed quadratic, whereas the loop counter should be running over the quadratics instead.

I also think Line 9 should read:

$Q^{(t)} \gets Q^{(t)} + w_i^{(t)}a_ia_i^{\top}$ (I think you need an outer product and not an inner product, same thing for Line 6)

I think you also need an averaging step at the very end, as the [CCLY19] algorithm returns the weights $w = 1/T \cdot \sum_{t'=1}^{T} w_{t'}$.

Finally, I think this algorithm actually can be implemented in $d^2T \le d^{\omega}T$ time per row. This is because of associativity:

$\prod_{t=1}^t a_i^{\top}Q_{t}^{-1}a_i = \prod_{t=1}^t a_i^{\top}(Q_{t}^{-1}a_i)$

Now, $(Q_{t}^{-1}a_i)$ is a vector that can be formed in time $d^2$ via naive multiplications, and then $a_i^{\top}(Q_{t}^{-1}a_i)$ is a dot product that can be found in $d$ time. We do this $t$ times for each term in the product and then multiply them all together for a total time of $d^2t$ for step $t$.

**Questions:**

See all of the above.

**Limitations:**

Yes

---

> ### Author Rebuttal · Authors · 2024-08-06
>
> > I wish more applications had been discussed. ...
>
> We note that $(1+\epsilon)$-approximate John ellipsoids have many applications to statistics, machine learning, and computational geometry, as is discussed in our introduction as well as in the works of [CCLY19, SYYZ22]. Some notable applications include D-optimal experiment design, outlier detection, and pattern recognition.
>
> There are a couple of difficulties which prevent us from stating further running time improvements as a result of our faster leverage score algorithm beyond our applications to John ellipsoids, and we believe our work highlights the importance of overcoming such difficulties. The first is that the target application should require $(1+\epsilon)$-approximate leverage scores, since if only constant factor approximations are required (as in the case of $\ell_p$ Lewis weight sampling), then leverage score approximation can be done in input sparsity time. If we wish to compute $(1+\epsilon)$-approximate $\ell_p$ Lewis weights using $(1+\epsilon)$-approximate leverage scores, then known reductions require either exact leverage scores [FLPS22] or leverage score approximations that are $(1+\epsilon/\mathrm{poly}(d))$-approximate [AGS24]. In the latter case, in theory our results may no longer give improvements due to the large exponent in our polynomial dependence on $\epsilon$. We have raised the question of whether faster algorithms can be design for $\ell_p$ Lewis weights in the original draft, and we will include this discussion in the revision.
>
> > Mild issues: ...
>
> Thank you for pointing these out! We have fixed these in the revision. The $d^\omega$ dependence (rather than the $d^2$ that you suggest) is due to computing the matrix inverse $Q^{-1}$.

---

> > ### Comment · Reviewer_VRzA · 2024-08-07
> > **responding to author response to review**
> >
> > Hi, thanks for the clarification! I have a couple of followup questions:
> >
> > > if only constant factor approximations are required (as in the case of $\ell_p$ Lewis weight sampling), then leverage score approximation can be done in input sparsity time
> >
> > I thought leverage scores for general inputs took $\widetilde{O}(nnz(A) + d^{\omega})$ time to compute (which looks like it could be worse than the runtime you get for dense matrices)? They only run in truly input sparsity time when $A$ has additional structure (e.g. if $A$ is a graph edge incidence matrix)? See, e.g., Lemmas 7-10 of https://arxiv.org/pdf/1408.5099. Of course, it's likely I misunderstood what you meant by input-sparsity time though.
> >
> > > The [runtime] is due to computing the matrix inverse
> >
> > Ah, I forgot to mention -- can you maintain $Q_{t}^{-1}$ using Sherman-Morrison (https://en.wikipedia.org/wiki/Sherman%E2%80%93Morrison_formula)? In particular, each update to $Q$ looks like a rank-$1$ update, so the formula for updating $Q^{-1}$ follows from Sherman-Morrison (and it looks like to me that each update to $Q^{-1}$ runs in $d^2$ time). The formula as written in the wiki page only applies to invertible matrices but I think it holds (https://mathoverflow.net/questions/146831/sherman-morrison-type-formula-for-moore-penrose-pseudoinverse) for maintaining the pseudoinverse as well if the matrix you are keeping track of symmetric -- which it looks like it is.
> >
> > Thanks again for following up!

---

> ### Author Response · Authors · 2024-08-08
>
> Indeed, constant factor leverage score approximation takes time $\tilde O(\mathrm{nnz}(A) + d^\omega)$ time, as we cite in Theorem 1.2. We have informally referred to this as input sparsity time, as we assume a regime where $\mathrm{nnz}(A) \geq n \gg \mathrm{poly}(d)$. For dense matrices, our $\tilde O(nd)$ dominating running time would upper bound $\mathrm{nnz}(A)$. To re-iterate, our running time improvement is focused on $(1+\epsilon)$ approximation of leverage scores, where we improve from $\epsilon^{-2}\mathrm{nnz}(A) + \mathrm{poly}(d/\epsilon)$ to $\tilde O(nd) + \mathrm{poly}(d/\epsilon)$. However, this does not give substantial improvements when $\epsilon$ is constant.
>
> Yes, we can use the Sherman--Morrison formula to improve the update time of maintaining the inverse of the quadratic in our streaming algorithm, thank you for pointing that out! It is unclear if the Sherman--Morrison would improve the running time of the offline fast algorithm.

---

> > ### Comment · Reviewer_VRzA · 2024-08-12
> > **response to author response to reviewer response to author response to review**
> >
> > OK, thanks a lot for checking the above! I have updated my confidence score accordingly.

---

### Official Review · Reviewer_hiPW · 2024-07-13

**Soundness:** 2
**Presentation:** 1
**Contribution:** 2
**Rating:** 3
**Confidence:** 2

**Summary:**

This paper considers the computing of an approximate John ellipsoid. They improve the algorithm by lazy update and fast matrix multiplication. They also give low-space streaming algorithms using similar ideas.

**Strengths:**

This paper improves John ellipsoid algorithm via lazy update and fast matrix multiplication from O(d^{\omega-1}) to O(d).

**Weaknesses:**

Major:
- This paper is badly written.
- The authors do not give justifications in the checklist.

Minor:
- Please check the capitalization in the references.
- can you explicitly give the space complexity in Theorem 1.8?

**Questions:**

See weakness

---

> ### Author Rebuttal · Authors · 2024-08-06
>
> > The authors do not give justifications in the checklist.
>
> We believe we have included a justification for the paper checklist whenever the list item warrants additional justifications. In particular, we give an in-depth discussion of the limitations of our work in Section 3. We are happy to give further justification on anything.
>
> > Please check the capitalization in the references.
>
> Upon a review of the citations, we have ensured that the names in the following titles are properly capitalized:
> - "An elementary proof of a theorem of Johnson and Lindenstrauss"
> - "Computing Lewis Weights to High Precision"
> - "On computing approximate Lewis weights"
> - "A near-optimal algorithm for approximating the John ellipsoid"
> - "Linear convergence of a modified Frank-Wolfe algorithm for computing minimum-volume enclosing ellipsoid"
> - "Faster Algorithm for Structured John Ellipsoid Computation"
>
> > can you explicitly give the space complexity in Theorem 1.8?
>
> The space complexity for Theorem 1.8 is included in the paragraph following Theorem 1.8, and the total space usage is $O(d^2 T)$ words (i.e., real numbers) of space, where $T = O(\epsilon^{-1}\log(n/d))$ is the number of passes. This has been clarified in the theorem statement of Theorem 1.8 in the revision.

---

### Decision · Program_Chairs · 2024-09-25

**Decision:**

Accept (poster)

**Comment:**

This paper improves upon the computational state of the art for computing John's ellipsoids. The focus is on giving low-storage algorithms amenable to streaming data setting, and is based on a simple idea of delaying leverage scores via sampling, and then using fast matrix multiplication in batches to speed up the bottleneck with high approximation accuracy. The main weaknesses in my view are that the paper is presented in a way that feels narrow in scope, and as one reviewer points out, it can come across as a fairly incremental contribution. However, the results are correct and carefully executed, and the paper can be further improved incorporating some comments from useful reviews such as further acceleration using Woodbury approximations. I would encourage the authors to give more context on how these contributions can allow users to tackle problems that may have been out of reach due to the previous bottleneck, and just widen the scope of the paper even though the contribution is technical in nature.

I recommend acceptance after studying the paper myself; due to the brevity and lack of substantive criticism in review hiPW, I chose to dismiss that score in the decision making process